# First Experiences with Ultrasound-Guided Transthoracic Needle Biopsy of Small Pulmonary Nodules Using One-Lung Flooding: A Brief Report

**DOI:** 10.3390/diagnostics15182374

**Published:** 2025-09-18

**Authors:** Thomas Lesser, Christian König, Seyed Masoud Mireskandari, Uwe Will, Frank Wolfram, Julia Gohlke

**Affiliations:** 1Department of Thoracic and Vascular Surgery, SRH Wald-Klinikum Gera, 07548 Gera, Germany; frank.wolfram@srh.de (F.W.); julia.gohlke@srh.de (J.G.); 2Department of Anesthesiology and Intensive Care, SRH Wald-Klinikum Gera, 07548 Gera, Germany; christian.koenig@srh.de; 3Institute of Pathology, SRH Wald-Klinikum Gera, 07548 Gera, Germany; seyed.mireskandari@srh.de; 4Department of Internal Medicine, SRH Wald-Klinikum Gera, 07548 Gera, Germany; uwe.will@srh.de

**Keywords:** indeterminate small pulmonary nodules, one-lung flooding, ultrasound-guided transthoracic core needle biopsy

## Abstract

**Introduction:** Non-surgical biopsy is recommended for diagnosing solid pulmonary nodules measuring >8 mm when the probability of malignancy is low to moderate. However, currently available biopsy methods do not have a sufficient diagnostic yield for nodule size <20 mm. Previous work has shown that one-lung flooding (OLF) enables complete lung sonography and good demarcation of lung nodules. Therefore, here, we report the first experiences with ultrasound-guided transthoracic core needle biopsy (USgTTcNB) under OLF for the histological diagnosis of small pulmonary nodules. **Methods:** In two patients with small pulmonary nodules, a transbronchial/thoracic biopsy was not indicated due to the size and location of the nodules. Following nodule detection under OLF, the USgTTcNB was performed. The biopsy cylinder was immediately examined via the frozen section procedure. After liquid draining and re-ventilation, the patients were extubated in the operation room and monitored in the intermediate care unit. **Results:** In both patients, a histological diagnosis was achieved. In the case of malignancy, the patient underwent lobectomy during the same session. In the case of a benign diagnosis, a futile operation was avoided. In case two, a small apical pneumothorax occurred. The hemodynamic values during and after the intervention were in the normal range. Lung function on day 2 after the intervention increased compared with that before the intervention. **Conclusions:** USgTTcNB under OLF is feasible and enables a histological confirmation of small pulmonary nodules. Nevertheless, this new promising technique should be evaluated in a study with a larger cohort.

## 1. Introduction

The tumor size in lung cancer determines the prognosis. At a size of 10 mm (T1b), the probability of five-year survival decreases from 92% to 84% [1]. Many guidelines recommend a non-surgical biopsy for solid nodules measuring >8 mm when the probability of malignancy is low to moderate [2,3,4]. The currently available biopsy methods do not have a sufficient diagnostic yield for a nodule size of <20 mm. Transbronchial lung biopsy using radial endobronchial ultrasound or electromagnetic navigation bronchoscopy with cone beam CT scanning has a diagnostic yield of 70% [5,6]. With virtual bronchoscopy navigation (NAVIGATOR-study), the diagnostic yields for lesions ≤2 cm were 37% [7], while CT-guided transthoracic biopsy shows a diagnostic yield of around 75% [8]. Therefore, many patients undergo surgery without prior diagnosis. Video- and robot-assisted thoracic surgeries are very successful when the nodules are located within 30 mm of the pleural surface [9]. In the case of deeper lesions, the risk of conversion to a thoracotomy increases. Furthermore, an unjustified high rate of segmentectomy or lobectomies occur without performing the frozen section procedure to clarify the diagnosis intraoperatively. Inevitably, patients undergo avoidable operations for benign lesions, with 9 to 35.6% of benign lesions being resected in the literature [10]. Therefore, with the increased detection of small indeterminate pulmonary nodules as a result of lung cancer screening, a non-surgical biopsy procedure with a high diagnostic yield is needed for small pulmonary nodules, involving not only the detection of malignancy but also a reliable benign diagnosis.

In previous experimental studies, we developed a method to completely visualize the lung via sonography. By filling one lung wing with isotonic saline (called one-lung flooding, OLF), the air content is replaced with fluid. With OLF, we have developed a technique for the percutaneous, transpleural sonographic detection and nodule punction of very small pulmonary nodules located deep in the parenchyma [11]. With two cases, we attempt to demonstrate the feasibility of ultrasound-guided transthoracic core needle biopsy (USgTTcNB) under OLF for the histological diagnosis of small pulmonary nodules.

## 2. Materials and Methods

We report on two patients who had indeterminate small pulmonary nodules with a moderate to high probability of malignancy according to the Mayo model [12]. In both cases, the interdisciplinary board recommended surgical biopsy because the success of both transbronchial and transthoracic biopsy was considered not promising. To avoid a diagnostic thoracotomy, USgTTcNB was suggested outside the guidelines. The study was conducted in accordance with the Declaration of Helsinki, and approved by the Ethics Committee LÄK Thüringen Germany (Approval no: 965/6139 and 895/6035, date: 20 June 2018 and 29 January 2025).

We thoroughly informed the patients about the new experimental method, which had not yet been used in humans. The patients agreed and provided written informed consent. In the case of non-diagnosed USgTTcNB, the patients also gave informed consent for a surgical biopsy, which would need to be performed in the same session. Each patient received body plethysmography with diffusing capacity, echocardiography, flexible bronchoscopy, and chest computer tomography (CT).

The patients were anesthetized and intubated with a left-sided double-lumen endobronchial tube (DLT), followed by total intravenous anesthesia and pressure-controlled ventilation of both lungs. A cuff controller (VBM Medizintechnik GmbH, Sulz a.N., Germany) was used to maintain a constant pressure of 50 cm H_2_O within both the endobronchial and tracheal cuffs. In the right lateral decubitus position, the right lung was filled (single filling) via the right tubus leg with degassed, warmed (37 °C) isotonic saline flowing passively from an infusion bottle suspended 30 cm above heart level (Appendix A).

After the right lung could be imaged with ultrasound (bk5000; bK medical GmbH, Quickborn, Germany), the patient was placed in supine position. To avoid dislocation of the DLT during this position change, the action was monitored via fiberbronchoscopy. After the pulmonary nodule was located, a biopsy needle was inserted under sonographic guidance, and a core biopsy was performed. The specimen was examined via the frozen section procedure. After that, the patients were placed in trendelenburg position to improve passive flow of the fluid through the opened right tubus leg. At the end of this drainage process, the tubus leg was connected to the ventilator, and both lungs were ventilated. After sufficient spontaneous breathing with an tcSO_2_ above 90%, the patients were extubated. The patients received an alternating oxygen mask (4 L/min) and continuous positive airway pressure (CPAP) and were monitored initially in the recovery room and later in the intermediate care unit. Then, 2 and 4 h after extubation, lung sonography was performed. After 6 h, a chest X-ray was performed. One week after discharge, ultrasound and X-ray follow-ups were performed.

The patients gave informed consent that in the case of malignancy, the tumor would be resected in the same session via videothoracoscopy or thoracotomy. In the case of a benign diagnosis, the patient would be extubated. A complete description of the method can be found in Appendix A.

## 3. Results

### 3.1. Case 1

A 65-year-old man presented to our lung cancer center for further evaluation of a solitary pulmonary nodule in the middle lobe, growing from 5 to 10 mm in size in two years (Appendix A). Four years ago, a rectum resection was performed due to adenocarcinoma. As described above, a USgTTcNB (14G, BARD^®^MAX-CORE^®^; BARD GmbH, Karlsruhe, Germany) under OLF was performed (Appendix A). With the first punction, an adenocarcinoma was histologically confirmed via the frozen section procedure. Since a differentiation between metastasis and lung carcinoma was not possible with the frozen section procedure, a lobectomy of the middle lobe was performed in the same session. The patient was extubated without any problems at the end of the intervention. Transthoracic ultrasound and chest X-ray showed neither pneumothorax nor pleural effusion. No postoperative complications occurred.

### 3.2. Case 2

Our second case was a 61-year-old patient suffering from a breast carcinoma on the right side. Twelve years ago, she received an operation and radio-chemotherapy due to a breast carcinoma on the left side. During staging, a chest CT-scan demonstrated a small pulmonary nodule about 5 mm in size in the middle lobe, which was new compared with a follow-up CT 6 months ago and suspected to be metastasis (Figure 1A). A histological confirmation was necessary for staging and to determine the therapeutic strategy. To avoid a thoracotomy, the patient was suggested a USgTTcNB under OLF. As described above, the right lung was filled with saline. A smoothly marginated hypoechoic nodule about 5 mm in size, which was directly adjacent to a vessel, was located 3 cm below the pleura (Figure 1B). Contrast-enhanced ultrasound revealed no perfusion into the nodule (Figure 1C). A USgTTcNB (18G, BARD^®^MAX-CORE^®^; BARD GmbH Karlsruhe, Germany) under OLF was performed (Figure 1D). The results from the frozen section procedure showed complete necrosis, confirmed after formalin fixation (Figure 1E). Since malignancy was extremely unlikely due to the 1 mm thick, 7 mm long core biopsy and no contrast enhancement occurred during ultrasound, the patient was extubated. Two hours later, a small apical pneumothorax was diagnosed via ultrasound and X-ray, and then evacuated using a thin (8F) pleural catheter, which could be removed after four hours. The further course of treatment was uncomplicated with normal blood gas analysis. Ultrasound and X-ray follow-ups on the second and ninth postoperative days showed no evidence of pleural effusion, pneumothorax, or residual fluid in the lungs (Appendix A). Body plethysmography on the second postinterventional day showed an increase in diffusing capacity and a decrease in residual volume and functional residual capacity compared with pre-intervention values (Figure 2). The patient was thus discharged on the third post-interventional day.

In both patients, hemodynamics and gas exchange were not affected during and after the procedure (Table 1). In the first two hours after extubation, ultrasound revealed residual fluid in the dorso-basal lung area, which was no longer detectable after 24 h.

## 4. Discussion

This work provides the first experiences with USgTTcNB of small pulmonary nodules under OLF. Using OLF, very small lung lesions could be localized with ultrasound deep in the lung parenchyma. In both cases, USgTTcNB was able to confirm the diagnoses for both a malignant and a benign nodule with one-time puncture. USgTTcNB could reduce the false-negative rate of benign, non-specific biopsies, which is currently between 7% and 29% [13]. In contrast to CT-guided TTNB, ultrasound enables the use of a real-time punction technique, which appears to increase the diagnostic yield, especially of small nodules. By using a core biopsy needle, sufficiently thick tissue cylinders are made available, thus allowing for histological examination in frozen sections. As case 2 shows, surgery can be avoided by confirming a benign diagnosis. On the other hand, in cases of malignancy, curative resection can be performed during the same session, which improves the patient’s comfort. Besides nodule localization and needle tracking, contrast-enhanced ultrasound can be used to distinguish benign from malignant lung lesions. Although in both patients, the nodules were surrounded by vessels, neither parenchymal nor pleural hemorrhages were observed. Ultrasound under OLF enables excellent visualization of bronchi and lung vessels, so injuries to these structures can be minimized. Furthermore, the positive alveolar pressure caused by the fluid filling supports hemostasis. Moreover, the reduced respiration motion during OLF is advantageous for precise ultrasound-guided puncture. However, a disadvantage of USgTTcNB under OLF is that in cases of non-diagnostic biopsy, an operation needs to consequently be performed. Furthermore, the method currently requires general anesthesia, DLT intubation, and one-lung ventilation. However, with further developments to the OLF procedure, such as selective filling of the lung lobe or segment-containing tumor, general anesthesia and intubation may be avoided.

OLF comprises a one-time filling with saline and is not comparable with lung lavage, in which intra-alveolar proteins are flushed out through numerous cycles of in- and outflow. Both of our patients had normal and stable hemodynamic values during and after the procedure. In case two, the intervention under OLF did not result in any deterioration in selected lung function values after the intervention. OLF not only enables USgTTcNB for histological confirmation but also could be suitable for the application of focused ultrasound for tumor ablation during the same session. Furthermore, an ultrasound-guided transbronchial biopsy under OLF is conceivable.

However, the following limitation should be considered: The feasibility and safety of USgTTcNB were shown on only two patients with normal lung function. Therefore, a larger cohort study should be performed, in particular in patients suffering from chronic obstructive pulmonary disease.

## 5. Conclusions

In conclusion, USgTTcNB under OLF is feasible and enables a histological confirmation of small pulmonary nodules via the frozen section procedure. By confirming a benign diagnosis, futile operations can be avoided. However, a larger study is needed to verify the diagnostic yield and safety of the procedure.

## Figures and Tables

**Figure 1 diagnostics-15-02374-f001:**
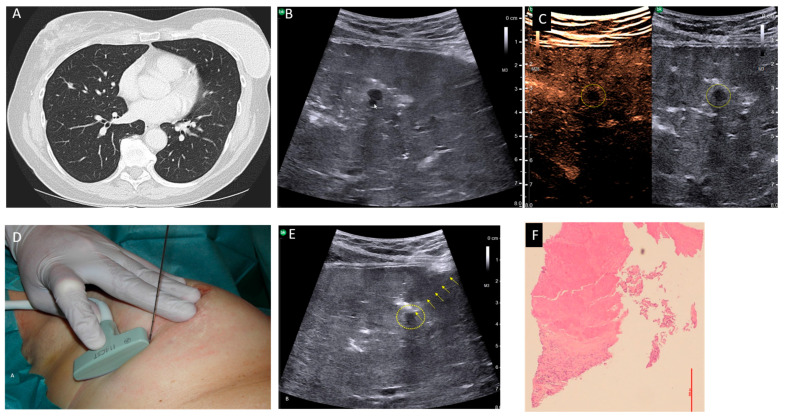
(**A**) CT scan showing a small pulmonary nodule adjacent to a vessel within the middle lobe. (**B**) Transthoracic ultrasound of the right lung under OLF revealing a smoothly marginated hypoechoic nodule adjacent to a vessel (arrow). (**C**) Contrast-enhanced ultrasound of the flooded lung showing a hypoechoic nodule (dashed circle) without contrast enhancement. (**D**) Free-hand ultrasound-guided biopsy using a T-shaped intraoperative transducer (bK medical GmbH, Quickborn, Germany). (**E**) Transthoracic lung ultrasound under OLF with a core biopsy needle (arrows) inserted into the nodule (dashed circle). (**F**) Histological image of the core biopsy showing a complete necrotic area above and vital normal lung tissue below (hematoxylin–eosin stain).

**Figure 2 diagnostics-15-02374-f002:**
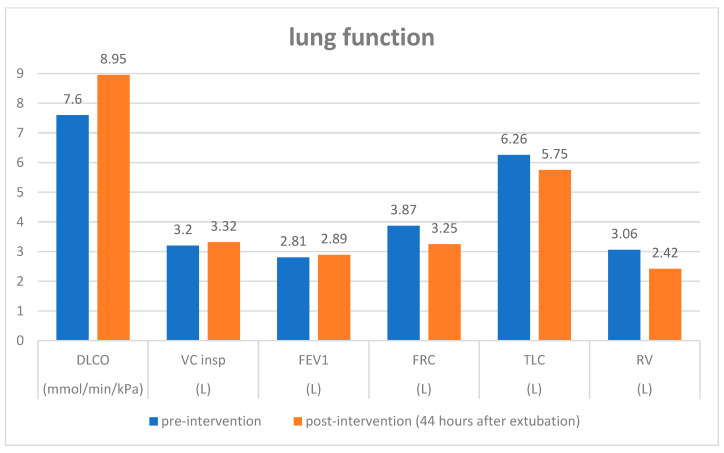
Lung function values of case 2 before and 44 h after USgTTcNB under OLF.

**Table 1 diagnostics-15-02374-t001:** The course of hemodynamic and gas exchange values of the two cases during and after one-lung flooding compared with pre-interventional values.

	Pre- Intervention		DLV		30 min After ELV + OLF		2 h After Extubation		12 h After Extubation	
Patient	C1	C2	C1	C2	C1	C2	C1	C2	C1	C2
HR	79	70	55	60	70	55	80	80	78	75
SAP (mmHg)	150	161	110	135	120	115	130	160	110	140
PaO_2_ (kPa)	10.7	11.80	41.1	46.7	26.2	31.4	8.6	9.9	10.5	12.8
PaCO_2_ (kPa)	5.1	4.5	5.3	6	4.3	4.8	5.1	5.4	4.9	4.9
tcSO_2_ (%)	95	97	99	99	95	99	94	94	96	97

DLV, double-lung ventilation; ELV, one-lung ventilation; OLF, one-lung flooding; HR, heart rate; SAP, systolic arterial pressure; PaO_2_, arterial partial pressure of oxygen; PaCO_2_, arterial partial pressure of carbon dioxide; tcSO_2_, transcutaneous oxygen saturation; C1, case 1; C2, case 2.

## Data Availability

The datasets generated or analyzed during this study can be made available upon request to the corresponding author.

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
