# Peer review of "First Experiences with Ultrasound-Guided Transthoracic Needle Biopsy of Small Pulmonary Nodules Using One-Lung Flooding: A Brief Report"

_diagnostics, 2025, doi:10.3390/diagnostics15182374_

Round 1
Reviewer 1 Report
Comments and Suggestions for Authors
Case selection. The woman with a 5 mm. nodule years after breast cancer treatment should not have been biopsied. Any reasonable tumor board would have recommended an interval CT scan to verify growth before biopsy. If this had been done, the patient would demonstrated no growth and been spared a major intervention.
This is clearly a research method and yet no mention is made of an IRB approval process or presentation to the patient of a research document.
The only articles found on "one lung flooding" in PubMed are all from Dr. Lesser. They are all animal and postmortem studies, with no prior use in humans.
The use of a method that requires the general anaesthesia with double lumen intubation and filling the lung with fluid, followed by hours on post procedure intubation and mechanical ventilation is a major intervention, compared with standard, awake CT guided transthoracic needle biopsy. Given the tiny series, biopsy of a benign nodule 50%, pneumothorax 50% are hardly examplary results.
Publication of this paper would be irresponsible. Provision of the article to hospital and public health agencies for investigation of ethical research practices would be appropriate, unless the authors provide clear documents showing that these procedures were done in an IRB approved research project and that the patient was suitably informed.
Reviewer 2 Report
Comments and Suggestions for Authors
The authors reported their experience of 2 cases with “USgTTcNB” for small pulmonary nodules under “OLF”. According to the authors, it is the first experience worldwide. As the authors described, the currently available biopsy methods do not have a sufficient diagnostic yield for small nodules. So it is very interesting report about new technique.
However, I think there are several problems in this paper.
Major point:
・According to the authors, it is the first trial in the world. In Line 69-70, they wrote that the patients gave written consent. However, they do not describe about institutional review board approval. Was this study performed following the right ethical manner? “Institutional review board statement” section from Line 213, the authors do not fulfill the required things. Is it the sentences from submission guidelines?
Minor points:
・In the case 2, the authors thought that “malignancy was ruled out” because the frozen section showed a complete necrosis, however, necrosis can be shown in the malignant lesions. Even if they detected specific pathogen like fungus or mycobacterium, they can be co-exist with cancer lesion. It is difficult to assert “non-malignant”.
・Although the result was “non-malignancy” and the patient was not resected the lung, 3 days hospital stay was required. Is it really less invasive method? How long do the authors make their patients stay after bronchoscopy or CT-guided needle biopsy?
・The authors said, “except for a small apical pneumothorax in case 2, no complications occurred”. However, the complication ratio was 50% in these 2 cases.
・In Line 164-5, the authors wrote, “USgTTcNB could reduce the false-negative rate of benign, non-specific biopsies”. However, they do not sufficiently describe the advantage of this method comparing to other biopsy methods. In Line 158-9, they wrote the advantage of “real-time puncture”. What else?
・The authors described the “increased” lung function after their procedure in the case 2 and explained that it is due to the “cleaning” of the small airways. However, I think they say too much. It is only 1 case experience. Moreover it is a repot about new biopsy method. If they perform this technique expecting the effect of air way cleaning, they should plan another study.
Round 2
Reviewer 2 Report
Comments and Suggestions for Authors
I think the authors reviesed the manuscript appropriately following to the former review.
I think it is acceptable now.